# Distribution of Extended-Spectrum β-Lactamase Genes and Antimicrobial Susceptibility among Residents in Geriatric Long-Term Care Facilities in Japan

**DOI:** 10.3390/antibiotics11010036

**Published:** 2021-12-29

**Authors:** Dai Akine, Teppei Sasahara, Kotaro Kiga, Ryusuke Ae, Koki Kosami, Akio Yoshimura, Yoshinari Kubota, Kazumasa Sasaki, Yumiko Kimura, Masanori Ogawa, Shinya Watanabe, Yuji Morisawa, Longzhu Cui

**Affiliations:** 1Division of Clinical Infectious Diseases, School of Medicine, Jichi Medical University, Yakushiji 3311-1, Shimotsuke 329-0498, Tochigi, Japan; m00001da@jichi.ac.jp (D.A.); yujim@jichi.ac.jp (Y.M.); 2Health Service Center, Jichi Medical University, Yakushiji 3311-1, Shimotsuke 329-0498, Tochigi, Japan; masa-oga@jichi.ac.jp; 3Division of Bacteriology, School of Medicine, Jichi Medical University, Yakushiji 3311-1, Shimotsuke 329-0498, Tochigi, Japan; k-kiga@jichi.ac.jp (K.K.); swatanabe@jichi.ac.jp (S.W.); longzhu@jichi.ac.jp (L.C.); 4Division of Public Health, Center for Community Medicine, Jichi Medical University, Yakushiji 3311-1, Shimotsuke 329-0498, Tochigi, Japan; k.kosami@jichi.ac.jp; 5Medical Corporation Sanikukai Nissin Hospital, Hishimachi 3 chome, Kiryu 376-0001, Gunma, Japan; yoshimuraakio@gmail.com; 6Nikko Citizen’s Hospital, Kiyotakiarasawamachi 1752-10, Nikko 321-1441, Tochigi, Japan; 97024yk@jichi.ac.jp; 7Clinical Microbiology Laboratory, Jichi Medical University Hospital, Yakushiji 3311-1, Shimotsuke 329-0498, Tochigi, Japan; kazumasa@omiya.jichi.ac.jp (K.S.); yumi-kimu@jichi.ac.jp (Y.K.)

**Keywords:** extended-spectrum β-lactamase, geriatric long-term care facilities, antimicrobial susceptibility

## Abstract

A high prevalence of extended-spectrum β-lactamase-producing Enterobacterales (ESBL-PE) may call for monitoring in geriatric long-term care facilities (g-LTCFs). We surveyed the distribution of ESBL-causative gene types and antimicrobial susceptibility in ESBL-PE strains from residents in g-LTCFs, and investigated the association between ESBL-causative gene types and antimicrobial susceptibility. First, we analyzed the types of ESBL-causative genes obtained from 141 ESBL-PE strains collected from the feces of residents in four Japanese g-LTCFs. Next, we determined the minimum inhibitory concentration values for alternative antimicrobial agents against ESBL-PE, including β-lactams and non-β-lactams. *Escherichia coli* accounted for 96% of the total ESBL-PE strains. Most strains (94%) contained *bla*_CTX-M_ group genes. The genes most commonly underlying resistance were of the *bla*_CTX-M-9_ and *bla*_CTX-M-1_ groups. Little difference was found in the distribution of ESBL-causative genes among the facilities; however, antimicrobial susceptibility differed widely among the facilities. No specific difference was found between antimicrobial susceptibility and the number of ESBL-causative genes. Our data showed that ESBL-PEs were susceptible to some antimicrobial agents, but the susceptibility largely differed among facilities. These findings suggest that each g-LTCF may require specific treatment strategies based on their own antibiogram. Investigations into drug resistance should be performed in g-LTCFs as well as acute medical facilities.

## 1. Introduction

The importance of health care management in geriatric long-term care facilities (g-LTCFs) is growing worldwide in countries with large aging populations. Infectious diseases continue to be major issues in g-LTCFs [1,2,3]. In particular, the spread of multidrug-resistant bacteria in g-LTCFs has recently become a serious problem [4,5,6]. The prevalence of extended-spectrum β-lactamase (ESBL)-producing Enterobacterales (ESBL-PE) has recently been noticeable in g-LTCFs in many countries [7,8,9,10,11,12,13,14,15,16]. These organisms are typically found in the intestinal and urinary tracts, but they often cause infections such as pneumonia, urinary tract infections, and bloodstream infections [17]. This might result in poor therapeutic outcomes and higher health care costs because of their resistance to many broad-spectrum antimicrobial agents [18]. Thus, extensive basic data on ESBL-PE are needed for effective infection control.

Epidemiological assessments for genotypes of ESBL-causative genes can be crucial in identifying the mechanism of ESBL-PE spread. This may also provide the basic evidence for considering effective infection control measures for ESBL-PE. However, trends in ESBL-causative gene types widely differ by country and region [19]. Although genetic studies on ESBL-PE in acute care hospitals have been previously carried out [20,21,22], only a few studies have focused on them in g-LTCFs.

Standard antimicrobial agents for infections caused by ESBL-PE are carbapenems [23]. However, the use of alternative agents for mild diseases was initially attempted because carbapenem overuse contributed to the emergence of carbapenem-resistant organisms in medical facilities [24]. In g-LTCFs as well, sparing carbapenems by using alternative antimicrobial agents may be advantageous for suppressing carbapenem-resistant organisms. However, because little information is available on the antimicrobial susceptibility of ESBL-PE strains to each alternative antimicrobial agent, there is no evidence that alternative antimicrobial agents can be safely used for empiric treatment targeting ESBL-PE infections among residents in g-LTCFs.

Inheritance of drug resistance genes among bacteria may accelerate multidrug resistance [25,26,27]. Furthermore, the acquisition of multiple drug resistance genes may be associated with decreased antimicrobial susceptibility [28]. However, little information is available on the association between the accumulation of ESBL-causative genes and antimicrobial susceptibility. Moreover, no studies have assessed whether monitoring ESBL-causative genes can help clinicians consider appropriate treatment strategies, although some commercial-based diagnostic tools for ESBL-causative genes have recently become available.

The primary purpose of this study was to survey the distribution of ESBL-causative genes and the antimicrobial susceptibility of ESBL-PE strains detected from residents in g-LTCFs in Japan. The second objective was to investigate the association between ESBL-causative gene types and antimicrobial susceptibility to assess potentially appropriate antimicrobial use in the initial treatment of infectious diseases due to ESBL-PE in g-LTCF residents.

## 2. Results

Table 1 shows the prevalence of ESBL-causative gene types (*bla*_TEM_, *bla*_SHV_, and *bla*_CTX-M_ group) in sample strains according to the facility. Because some strains have multiple ESBL-causative gene types, the total number of genes exceeds the total number of strains investigated (*n* = 141), resulting in *n* = 181 genes. The “*bla*_CTX-M-9_ group” was the most common (48.6%, 88/181), followed by the “*bla*_CTX-M-1_ group” (24.3%, 44/181) and “*bla*_TEM_” (17.7%, 32/181). Meanwhile, “*bla*_SHV_”, “*bla*_CTX-M-2_ group”, and “*bla*_CTX-M-8_ group” genes were relatively uncommon, at 6.6%, 2.2%, and 0.6%, respectively. When calculating percentages using the participant number (*n* = 141) as the denominator, the prevalence was 62.4% (88/141) for “*bla*_CTX-M-9_ group” and 31.2% (44/141) for “*bla*_CTX-M-1_ group”. There were no statistically significant differences among facilities for these ESBL-causative gene types except for *bla*_SHV_.

The prevalence of combinations of ESBL-causative genes is shown in Table 2. A total of 104 strains (73.8%) carried a single ESBL-causative gene type, while 37 strains (26.2%) had multiple types. The number of strains containing *bla*_CTX-M_ group genes was 133 (94%); only 8 strains (5.7%) did not contain *bla*_CTX-M_ group genes. The most frequent patterns were “*bla*_CTX-M-9_ group only” (51.1%), followed by “*bla*_CTX-M-1_ group only” (14.9%), and “*bla*_CTX-M-1_ group + *bla*_TEM_” (8.5%).

The antimicrobial susceptibility (%) of alternative agents against ESBL-PE according to the facility are shown in Table 3. Among antimicrobial agents, the overall susceptibility for cefmetazole (CMZ), flomoxef (FMOX), tazobactam/piperacillin (TAZ/PIPC), ceftolozane–tazobactam (TAZ/CTLZ), and fosfomycin (FOM) exceeded 80%; FMOX had the best susceptibility, 97%, followed by FOM and CMZ (both exceeding 90%). Ceftazidime (CAZ), CMZ, TAZ/PIPC, gentamicin (GM), and FOM significantly differed in antimicrobial susceptibility among the facilities.

Table 4 shows the antimicrobial susceptibility of ESBL-PEs to β-lactam agents by the number of ESBL-causative genes (single or multiple). No statistical significance was found in antimicrobial susceptibilities when classified by single or multiple ESBL-causative gene types.

## 3. Discussion

We analyzed 141 strains of ESBL-PE carried by residents in four g-LTCFs in Japan and investigated the distribution of ESBL-causative gene types and antimicrobial susceptibility. Among the ESBL-PE strains, *E. coli* accounted for 96% of the total, and the *bla*_CTX-M-9_ group and *bla*_CTX-M-1_ group were the most common ESBL-causative gene types, which is consistent with previous studies conducted in community or clinical settings in Japan [29,30,31,32,33]. With a few exceptions, there was no significant institutional difference in the distribution of ESBL-causative gene types, suggesting that the spread of the *bla*_CTX-M_ group is not a problem in a single g-LTCF or a region but might be a nationwide issue. Despite little difference in the distribution of ESBL-causative gene types, antimicrobial susceptibility differed significantly among g-LTCFs. In this study, although strains containing multiple ESBL-causative gene types accounted for 26% of the total strains, antimicrobial susceptibility did not significantly change as the number of ESBL-causative genes increased. Our data indicate that each g-LTCF may require specific treatment strategies based on their own antibiogram. Investigations into drug resistance should be performed in g-LTCFs as well as acute medical facilities.

Although carbapenems are the first-line drugs for treating ESBL-PE infections [34], appropriate use should be carefully considered because of the increased number of carbapenem-resistant Gram-negative bacteria [35]. Therefore, a carbapenem-sparing strategy in the treatment of mild to moderate infections caused by ESBL-PE should be required. CMZ is one of the most promising alternatives to carbapenems because, in theory, cephamycins cannot be degraded by ESBL-PE. Some studies have suggested that cephamycins do not differ from carbapenems in terms of treatment outcomes [36,37,38]. In our study, ESBL-PE strains from some g-LTCFs showed low antimicrobial susceptibility to CMZ. Therefore, if CMZ is to be chosen as an empiric therapy, the antimicrobial susceptibility of each facility should be ascertained beforehand. FMOX, belonging to oxacephems, is available and often used in Japan, Taiwan, and China. FMOX is a synthetic antimicrobial agent in which sulfur is substituted by oxygen in the basic skeleton of 7-ACA, which differs from the structure of CMZ. Despite the structural differences, FMOX is often used as an alternative to CMZ. Along with CMZ, FMOX is being examined as a potential alternative to carbapenem in the treatment of ESBL-PE infections [39]. In our study, FMOX susceptibility was high, suggesting the potential of FMOX in treating ESBL-PE infections. However, FMOX use should be avoided in severe infections because of its higher mortality rates than carbapenem use for treating bloodstream infections caused by ESBL-PE [39,40,41]. FOM inhibits phosphoenolpyruvate transferase, the first enzyme involved in the synthesis of peptidoglycan, thereby inhibiting cell-wall synthesis [34,42]. Because it does not have a β-lactam ring, it is expected to be effective in the treatment of infections caused by ESBL-PE. In fact, our results showed good susceptibility. However, Babiker et al. [43] reported a clinical success rate of only 56% for FOM in treating urinary tract infections caused by ESBL-PE. Another previous study indicated the superiority of FOM to carbapenems in the treatment of urinary tract infections with ESBL-PE; however, these results have not yet been made available to the public [44]. TAZ/CTLZ, a new β-lactam/β-lactamase inhibitor, also shows promise as a treatment option for ESBL-PE infections; it was found to be equally as successful as levofloxacin (LVFX) for urinary tract infections, as well as carbapenems for intra-abdominal infections [45]. Although clinical use of TAZ/CTLZ was only initiated in 2019 in Japan, it has a resistance rate of approximately 20% among ESBL-PE strains in one g-LTCF of our study. This highlights the urgency of addressing this situation.

This study has some limitations. We only focused on ESBL-causative genes, potentially resulting in introducing bias to our results, which was a primary limitation of this study. The gene examination kit used in this study can only analyze a group of ESBL-causative genes. Multiple β-lactamases (i.e., ESBL and AmpC) may coexist in our samples [46,47,48,49]. Resistance mechanisms other than ESBL production might affect antimicrobial susceptibility. For more detailed characterization, other testing methods such as whole-genome sequencing should be used. Second, this study was limited to four LTCFs in three regions in Japan. For a wider, generalized picture, a larger number of g-LTCFs across Japan should be investigated. Finally, we investigated antimicrobial susceptibility using an automated instrument for infectious disease testing. Although this machine is often used in clinical practice, the accuracy of its MIC might have some variation.

## 4. Materials and Methods

### 4.1. Sample Strains

We conducted a cohort study among residents receiving long-term care in four g-LTCFs in Japan. The facilities were selected because they had secure linkage with their own specific backup hospitals where residents were typically transferred for any medical needs. All four g-LTCFs were anonymized in accordance with the ethics protocol of this study. Two related studies have been published using the same study settings and participants as the present study [50,51]. The Jichi Medical University Bioethics Committee for Medical Research approved this study and waived the requirement for informed consent from individual participants (Approval ID: 21–106).

In Japan, g-LTCFs are classified into two main types: (1) geriatric health service facilities and (2) geriatric special nursing homes. The former are intermediate facilities between hospitals and nursing homes, with a primary focus on rehabilitation. These facilities typically aim to return patients to home-based care, although some residents may require long-term care for years. The latter provide daily life support, including end-of-life care [50,51,52,53]. This study included three geriatric health service facilities and one geriatric special nursing home.

The study participants were 260 older adults residing in the four g-LTCFs who underwent ESBL-PE carriage testing during the study period (August 2018 through March 2020) [52]. Residents who did not undergo ESBL-PE testing were excluded from this study. Background data for the 260 residents, such as age, sex, and general condition, were not obtained because of the ethics protocol employed in this study.

We previously documented microbiology and ESBL-PE isolate procedures in a related study [51]. Small amounts (0.05–0.10 g) of freshly voided feces from the residents were obtained from paper diapers or papers for self-collection of stool samples (AS ONE Corporation, Osaka, Japan) using stool collection tubes with agar medium (FECES COLLECTING TUBE™; Eiken Chemical Co. Ltd., Tokyo, Japan). To avoid contamination, facility staff performed standard preventive measures during sample collection.

Each fecal sample was directly harvested on selective screening agar plates for ESBL-PE (CHROMagar™ESBL/CHROMagar™ mSuperCARBA bi-plate medium [Kanto Chemical Co. Inc., Tokyo, Japan]) using sterile cotton swabs. This medium was able to grow ESBL-PE simultaneously with carbapenem-resistant Enterobacterales and/or AmpC β-lactamase-producing bacteria. Following aerobic incubation at 37 °C for 18–24 h, colonies growing on either side of the adapted screening agar were regarded as suspect for any of the above bacteria, and underwent further analysis. ESBL-PE and AmpC β-lactamase-producing bacteria were identified using an ESBL + AmpC Detection Set™ (MAST Group, Bootle, Merseyside, UK).

Bacterial species and susceptibilities were identified by the Vitek™ 2 automated instrument (bioMérieux); these species were subsequently confirmed using Matrix-Assisted Laser Desorption/Ionization Time-of-Flight Mass Spectrometry (MALDI-TOF-MS; Bruker Daltonics Inc., Bremen, Germany).

ESBL production was confirmed in all isolates by combination disk diffusion testing according to Clinical and Laboratory Standards Institute (CLSI) guidelines, using cefotaxime and CAZ as indicator molecules alone and in combination with clavulanic acid [54]. Results were considered positive when the zone diameter around the disk containing the drug in combination with clavulanic acid was ≥5 mm larger than that around the disk containing the drug alone.

### 4.2. Identification of ESBL-PE Strains (Polymerase Chain Reaction Amplification)

We analyzed genotypes of ESBL-causative genes for all strains. DNA isolation from ESBL-PE was performed using the Cica Geneus^TM^ DNA Extraction Kit (Kanto Chemical Co., Inc., Tokyo, Japan). Each colony that developed on sheep blood agar medium was transferred to DNA extraction reagents and incubated at 72 °C for 6 min and 94 °C for 3 min using a T100^TM^ Thermal Cycler (Bio-Rad Laboratories, Inc., Hercules, CA, USA) [55]. PCR was performed using the Cica Geneus^TM^ ESBL Genotype Detection Kit2 (Kanto Chemical, Tokyo, Japan) according to the manufacturer’s instructions. PCR to detect *bla*_TEM_, *bla*_SHV_, *bla*_GES_, *bla*_CTX-M-1_ group, *bla*_CTX-M-2_ group, *bla*_CTX-M-8_ group, *bla*_CTX-M-9_ group, *bla*_CTX-M-25_ group, and *bla*_CTX-M-chimera_ group genes was performed using the T100^TM^ Thermal Cycler with the following reaction profile: 30 cycles of 94 °C for 15 s, 63 °C for 15 s, and 72 °C for 40 s [56]. Electrophoresis was performed at 100 V for approximately 20 min with a 2% (wt/vol) agarose gel containing Atlas ClearSight DNA Stain (BioAtlas, Tartu, Estonia). The products were visualized under UV using a gel documentation system (AE-6933FXES-U Printgraph; ATTO, TOKYO, Japan). Using the DNA contained in the Cica Geneus^TM^ ESBL Genotype Detection Kit2, it was confirmed that the sequence of the target gene was amplified in the PCR reaction and the band size in DNA electrophoresis matched the amplified size of the target gene. In addition, as a negative control for ESBL gene detection, PCR was performed on a sample without DNA, and it was confirmed that no specific band was detected by DNA electrophoresis.

### 4.3. Susceptibility Testing for ESBL-PE Strains

We measured the minimum inhibitory concentration (MIC) for alternative antimicrobial agents against ESBL-PE, including β-lactam agents (cefepime, cephamycins, TAZ/PIPC, and TAZ/CTLZ) and non-β-lactams (aminoglycosides, quinolones, and FOM). Antimicrobial susceptibility testing was performed using an automated instrument for infectious disease testing (VITEK^TM^ 2 version 9.02; bioMérieux S.A.) with AST-N227, AST-N269, and AST-XN17 cards for the system. Antimicrobial susceptibility was determined mainly based on CLSI criteria (M100Ed30) [56]. Since there was no available breakpoint for FMOX, the CLSI breakpoints for latamoxef [susceptible (S) ≤ 8 μg/mL, resistant (R) ≥ 64 μg/mL for Enterobacterales] were used for FMOX [57,58]. For each antimicrobial agent, susceptibility is presented as a percentage of susceptibility-positive strains among all strains that underwent testing.

### 4.4. Analysis of the Relationship between Genetic Characteristics and Antimicrobial Susceptibility in ESBL-PE Strains

We investigated the difference between the number of ESBL-causative genes (single or multiple genes) and susceptibility to β-lactam agents because the accumulated number of ESBL-causative genes responsible for antimicrobial resistance mechanisms might be associated with decreased susceptibility to β-lactam agents [59].

### 4.5. Statistical Analysis

This study did not require sample size calculation since an existing database was used [60]. All analyses were performed using IBM SPSS Statistics for Windows, version 25 (IBM Corporation, Armonk, NY, USA). All categorical variables are presented as a percentage of patients. For the comparison of different groups, chi-square tests were used with a significance threshold at *p* < 0.05.

## 5. Conclusions

There was little difference in the distribution of ESBL-causative genes among the facilities. No specific association was found between the number of ESBL-causative genes and antimicrobial susceptibility. Although CMZ, FMOX, TAZ/CTLZ, and FOM can be identified as potential alternatives to carbapenems, each g-LTCF might have large variations in sensitivity to these antimicrobial agents. These findings suggest that each g-LTCF may require specific treatment strategies based on their own antibiogram. Investigations into drug resistance should be performed in g-LTCFs.

## Figures and Tables

**Table 1 antibiotics-11-00036-t001:** Prevalence of ESBL-causative genes: *bla*_TEM_, *bla*_SHV_, and *bla*_CTX-M_ group.

Gene Types	Total (*n* = 181)	Facility A (*n* = 21)	Facility B (*n* = 44)	Facility C (*n* = 38)	Facility D (*n* = 78)	*p*-Value ^†^
*bla* _TEM_	32 (17.7%)	1 (4.8%)	8 (18.2%)	6 (15.8%)	17 (21.8%)	0.27
*bla* _SHV_	12 (6.6%)	3 (14.3%)	0 (0%)	5 (13.2%)	4 (5.1%)	0.02 *
*bla*_CTX-M-1_ group	44 (24.3%)	5 (23.8%)	10 (22.7%)	9 (23.7%)	20 (25.6%)	0.74
*bla*_CTX-M-2_ group	4 (2.2%)	0 (0%)	2 (4.5%)	1 (2.6%)	1 (1.3%)	0.71
*bla*_CTX-M-8_ group	1 (0.6%)	0 (0%)	1 (2.3%)	0 (0%)	0 (0%)	0.48
*bla*_CTX-M-9_ group	88 (48.6%)	12 (57.1%)	23 (52.3%)	17 (44.7%)	36 (46.2%)	0.44

* Statistically significant; ^†^ chi-square tests.

**Table 2 antibiotics-11-00036-t002:** Prevalence of combinations of ESBL-causative genes.

Combination of Genes	Total *n* = 141	Facility A *n* = 15	Facility B *n* = 41	Facility C *n* = 27	Facility D *n* = 58	*p*-Value ^†^
*bla*_CTX-M-9_ group only	72 (51.1%)	9 (60%)	21 (51.2%)	14 (51.9%)	28 (48.3%)	0.88
*bla*_CTX-M-1_ group only	21 (14.9%)	―	8 (19.5%)	3 (11.1%)	10 (17.2%)	0.28
*bla*_CTX-M-1_ group + *bla*_TEM_	12 (8.5%)	1 (6.7%)	1 (2.4%)	2 (7.4%)	8 (13.8%)	0.25
*bla*_CTX-M-9_ group + *bla*_TEM_	9 (6.4%)	―	1 (2.4%)	1 (3.7%)	7 (12.1%)	0.23
*bla*_TEM_ only	6 (4.3%)	―	6 (14.6%)	―	―	<0.01 *
*bla*_CTX-M-1_ group + *bla*_CTX-M-9_ group	4 (2.8%)	2 (13.3%)	1 (2.4%)	1 (3.7%)	―	0.05
*bla*_CTX-M-1_ group + *bla*_SHV_	4 (2.8%)	2 (13.3%)	―	1 (3.7%)	1 (1.7%)	0.06
*bla*_CTX-M-2_ group only	4 (2.8%)	―	2 (4.9%)	1 (3.7%)	1 (1.7%)	0.71
*bla*_CTX-M-1_ group + *bla*_TEM_ + *bla*_SHV_	3 (2.1%)	―	―	2 (7.4%)	1 (1.7%)	0.18
*bla*_CTX-M-9_ group + *bla*_SHV_	3 (2.1%)	1 (6.7%)	―	1 (3.7%)	1 (1.7%)	0.43
*bla*_TEM_ + *bla*_SHV_	2 (1.4%)	―	―	1 (3.7%)	1 (1.7%)	0.60
*bla*_CTX-M-8_ group only	1 (0.7%)	―	1 (2.4%)	―	―	0.48

* Statistically significant; ^†^ chi-square tests.

**Table 3 antibiotics-11-00036-t003:** Prevalence of strains susceptible to antimicrobial agents by facility [%, (number of susceptible strains)].

Antimicrobial Agents	Total (*n* = 141)	Facility A (*n* = 15)	Facility B (*n* = 41)	Facility C (*n* = 27)	Facility D (*n* = 58)	*p*-Value ^†^
CAZ	65% (91)	73% (11)	63% (26)	89% (24)	51% (30)	<0.01 *
CFPM	61% (86)	73% (11)	61% (25)	70% (19)	53% (31)	0.34
CMZ	90% (127)	100% (15)	100% (41)	100% (27)	76% (44)	<0.01 *
FMOX	97% (137)	100% (15)	100% (41)	100% (27)	93% (33)	0.12
TAZ/PIPC	83% (117)	73% (11)	98% (40)	85% (23)	74% (43)	0.02 *
TAZ/CTLZ	91% (128)	80% (12)	95% (39)	93% (25)	90% (52)	0.36
GM	86% (122)	93% (14)	70% (29)	92% (25)	93% (54)	<0.01 *
LVFX	3.5% (5)	0% (0)	2.4% (1)	7.4% (2)	3.5% (2)	0.60
FOM	91% (128)	100% (15)	98% (40)	100% (27)	79% (46)	<0.01 *

CAZ, ceftazidime; CFPM, cefepime; CMZ, cefmetazole; FMOX, flomoxef; TAZ/PIPC, tazobactam/piperacillin; TAZ/CTLZ, tazobactam/ceftolozane; GM, gentamicin; LVFX, levofloxacin; FOM, Fosfomycin. * Statistically significant; ^†^ chi-square tests.

**Table 4 antibiotics-11-00036-t004:** Antimicrobial susceptibility to β-lactam antimicrobial agents by the number of ESBL-causative genes [%, (number of susceptible strains)].

Antimicrobial Agents	Single Gene (*n* = 104)	Multiple Genes * (*n* = 37)	*p*-Value ^†^
CMZ	88% (92)	95% (35)	0.28
FMOX	97% (101)	97% (36)	0.95
CAZ	63% (66)	68% (25)	0.65
CFPM	60% (62)	65% (24)	0.57
TAZ/PIPC	85% (88)	78% (29)	0.39
TAZ/CTLZ	91% (95)	89% (33)	0.70

CAZ, ceftazidime; CFPM, cefepime; CMZ, cefmetazole; FMOX, flomoxef; TAZ/PIPC, tazobactam/piperacillin; TAZ/CTLZ, tazobactam/ceftolozane. * Two or three ESBL-causative genes; ^†^ chi-square tests.

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
