# Peer review of "Distribution of Extended-Spectrum β-Lactamase Genes and Antimicrobial Susceptibility among Residents in Geriatric Long-Term Care Facilities in Japan"

_antibiotics, 2021, doi:10.3390/antibiotics11010036_

Round 1

Reviewer 1 Report

Enterobacterales having ESBL characteristics are major health concern. The manuscript describes the distribution of various ESBL genes in Enterobacterales and their resistance profile in residents staying in the geriatric long-term care facilities in Japan. Detection of ESBL genes were conformed by molecular based approach, in addition, MIC was also done to reveal the resistance profile. This is a well-designed simple study, but nice work and findings are interesting. My comments are as follows.

Please provide ethical permission of this study.

The sample size 141 is a small sample size to determine Prevalence (As shown in Table 4, line 123]. Can this sample size umber be justified? Please do a sample size calculation using the mathematical formula…to justify this.

Provide some info., on how those Escherichia coli, 135; Klebsiella pneumonia, 3; Proteus 217 mirabilis, 1; Citrobacter freundii, 1; Enterobacter cloacae, 1 were identified in the first instance. Is there any published work on these isolates, if so, please refer to those.

Line 231. No need for using the term Experiment …so, please Change the heading Experiment 1: Identification of ESBL-PE strains by polymerase chain reaction amplification TO “Identification of ESBL-PE strains by polymerase chain reaction”

Line 247. Also Experiment 2: Susceptibility testing for ESBL-PE strains

 To : “Susceptibility testing for ESBL-PE strains

And

Line 258. Experiment 3: Analysis of the relationship between genetic characteristics and antimicrobial susceptibility in ESBL-PE strains

 To :”Analysis of the relationship between genetic characteristics and antimicrobial susceptibility in ESBL-PE strains “

What were the positive and negative control in the PCR, write abbot it. Did any random sequencing was done to conform the  detection of ESBL gene in PCR??

Please give some management information of geriatric health services facilities and one geriatric special nursing home. Is there any major difference in the management system??

Line 203: Therefore, it is difficult to find a strategy for selecting appropriate antimicrobial agents based on the number of causative genes of ESBLs alone. …is there no evidence in this regard in literature??

Reviewer 2 Report

This is a cross-sectional study in Japan of ESBL-producing Enterobacterales taken from stool samples of residents at four long-term care facilities in Japan. The authors tried to survey the genotypes of ESBL genes, and association between phenotypic antibiotic resistance patten and genotypes. Among 141 strains with almost all of them were E. coli, they found that blaCTX-M group was by far the most common genotype. This study was unique in targeting residents in long-term care facilities as previous studies were mostly conducted in acute-care hospitals. While the first aim was important for local epidemiology, the second aim of their study was of unclear benefit for two reasons: 1. Phenotypic resistance is not solely determined by the presence or types of ESBL genes 2. Clinicians may consider the presence of ESBL in treatment, but not about the type of ESBL gene. Authors should clearly explain in background section why investigating the association between genotypes and phenotypes are important, and how could the results be used clinically. They should also discuss the limitation of not checking other resistant mechanisms.

My other specific comments are below,

  1. Table 2 – 3rd row, please correct 12(8%.5)
  2. Line 108 and in other places throughout the manuscript – check the spell of “Ceftrozan”. I think it should be “ceftolozane”.
  3. Table 4 – I do not think it meaningful to compare blaCTM – containing group versus lacking group because only 8 patients were in the lacking group.
  4. Table 5 – similar to table 4, comparing “three genes group” with only 3 patients would not be very meaningful. I suggest to combine latter two and compare “one gene” versus “two or more” to increase the power.
  5. Discussion – lines 137-144. This paragraph includes only general topic and should be incorporated in background section.
  6. Discussion - Lines 164-166, and 173-175. How the authors defined “poor” and “satisfactory” susceptibility rate? It is very subjective and depends on the clinical settings. I suggest to avoid subjective expressions and just discuss the objective numbers.
  7. Discussion – What is the possible explanations of not seeing worsening phenotypic resistance in strains with two or three ESBL genes compared to those with only one ESBL genes? Authors should make a discussion if they conducted the analysis.
  8. Methods – authors should explain more about how strains were collected. Are they used as a screening? Or clinical specimen collected for diagnosis? Or purely collected for study purpose?
  9. Methods – information about ethics was not included. Please include how ethical issues were addressed in this study.
  10. Methods – line 272. Please correct P<0.05.5.
  11. Conclusion – Lines 275-276. I think 50% of blaCTX-M-9 and 25% of blaCTX-M-1 were when denominator was 181 (numbers of all genes). But I guess denominator should be the number of strains (n=141). Please double check.

Round 2

Reviewer 2 Report

In this revised manuscript, authors appropriately answered my comments point by point. I feel the revised manuscript is ready for publication.